# Beijing Resident's Preferences of Ecosystem Services of Urban Forests

**Han Zhi-Ying [1] and Youn Yeo-Chang [1,2,\*]**

1 Department of Agriculture, Forestry, and Bioresources, College of Agriculture and Life Sciences, Seoul National University, 1 Gwanakro, Gwanakgu, Seoul 08826, Korea; hanzhiying@snu.ac.kr
2 Research Institute of Agriculture and Life Sciences, Seoul National University, 1 Gwanakro, Gwanakgu, Seoul 08826, Korea
\* Correspondence: youn@snu.ac.kr; Tel.: +82-02-880-4754

**Abstract:** This paper aims to investigate the Beijing resident's preferences over various options of urban forest management strategies. The literature investigation and expert Delphi method were conducted to classify the ecosystem services of urban forests into six categories: (1) fresh water provision, (2) noise reduction, (3) moderation of extreme events, (4) air quality regulation, (5) species diversity and wildlife habitat, and (6) recreation and spiritual experience. To elicit the relative preferences to ecosystem service (hereafter referred to as ES) of Beijing residents, we employed the choice experiment method. The data were collected by interviews with questionnaires conducted in October 2017, and a total of 483 valid questionnaires were analyzed. The subjects of this experiment were residents older than 19 years old who have lived in Beijing for more than 1 year and have visited any one of the urban forests located in Beijing more than once during 2016. The results were as follows: Firstly, the air quality regulation ES was considered as the most important service for Beijing residents in terms of their choices of urban forest. In addition, Beijing residents regarded the fresh water provision ES as the second most important ES. Beijing residents were willing to pay up to 1.84% of the average monthly income of Chinese households annually to expand urban forest ecosystems in order to improve air quality. Secondly, apartment owners were willing to pay more municipality tax for forest ESs than residents who did not own an apartment. Thirdly, residents were more willing to pay for urban forest ESs as their income increases. The results indicated that Beijing residents were willing to pay more tax in support of urban forestry for air quality improvement. This research suggests that urban environmental policy makers in Beijing should pay more attention to the regulation function of forests (especially improving air quality) when designing and managing urban forests.

**Keywords:** urban forest; ecosystem service; resident's preferences; choice experiment; Beijing

## 1. Introduction

### 1.1. Research Background

Natural urban ecosystems contribute to public health and improve the quality of life of urban residents [1,2]. The world's urban population has undergone significant growth. It has increased from 746 million in 1950 to 3.9 billion in 2014 [3]. By 2016, the proportion of this urban population is 54.37% of the total population of the world [4]. DeFries [5] mentioned that the world's population growth rates are slow, but urban growth is far overtaking rural growth. China is the most populous country in the world. Rapid economic development has resulted in dramatic changes in its urban population. Since China's reform and opening up, it has experienced the largest urbanization in the history of the world, with its urbanization level rising rapidly from 17.9% in 1978 to 56.1% in 2015 [6]. Rapid urbanization creates tremendous pressure on the natural environment. It also causes many ecological problems, especially in the city and surrounding areas [7].

Beijing is a mega city with a population of 21.536 million as of 2019 [8] with little green space, as seen in Figure 1. It has a typically continental monsoon climate with four different seasons. Most of its precipitation is concentrated in July and August. Half the year is a frost-free period [9]. In 2015, Beijing's GDP was 2.29686 trillion RMB. Calculated based on its resident population, its per capita GDP reached 106,284 RMB. In 2016, the per capita disposable income in Beijing reached 52,530 RMB, and the total retail sales of social consumer goods reached 1.10051 trillion RMB [10]. There were 25 smog days in Beijing during January 2013. The average visibility was 9.2 km, and the PM2.5 measurement reached a level of more than 800 $\mu g/m^3$. The number of smog days was 2.2 times more than during the same period of an ordinary year (11.4 days). This was the most severe pollution level since 1954 [11]. There have been frequent occurrences of smoggy weather in China since January 2013. This has become a pressing issue for the general public. Beijing, due to its unique natural situation and socioeconomic background, has become one of China's inland areas seriously affected by atmospheric pollution [12]. At the end of 2015, Beijing was attacked by heavy smog and announced two consecutive red alerts [13]. Beijing's air quality deteriorated. The smog not only affected people's health and quality of life but also had a negative effect on tourism [14]. Nowak [15] demonstrated that urban tree management could provide an effective way to improve urban air quality in the United States. Urban forest plays an essential role in creating an ecological environment, promoting capital growth, especially in terms of the maintenance of ecological security, and addressing climate change.

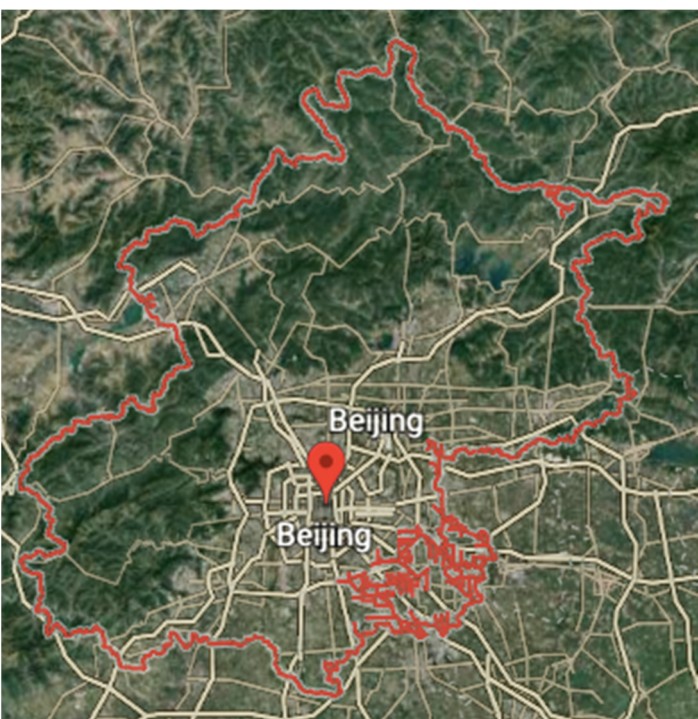

**Figure 1.** The map of Beijing (source: Google).

Comparing the forest coverage rate of other big cities around the world in 2012, Beijing has a low forest coverage rate with 14.85% (Tokyo (37.80%), London (34.80%), Paris (65.00%), New York (24.00%)) [16]. The World Health Organization (WHO) has recommended that an "international city is a healthy city, where its green space area per capita is 40–60 square meters and its park green space area per capita is 20 square meters". Compared with these requirements, Beijing is still lacking in this regard, as it has a low urban park green space per capita with 16 square meters [17]. Figure 2 shows the green spaces and urban forests distribution in Beijing in 2015. Beijing's high population density

and limited urban green space area stress the necessity that should attach importance to the design and management of urban forest.

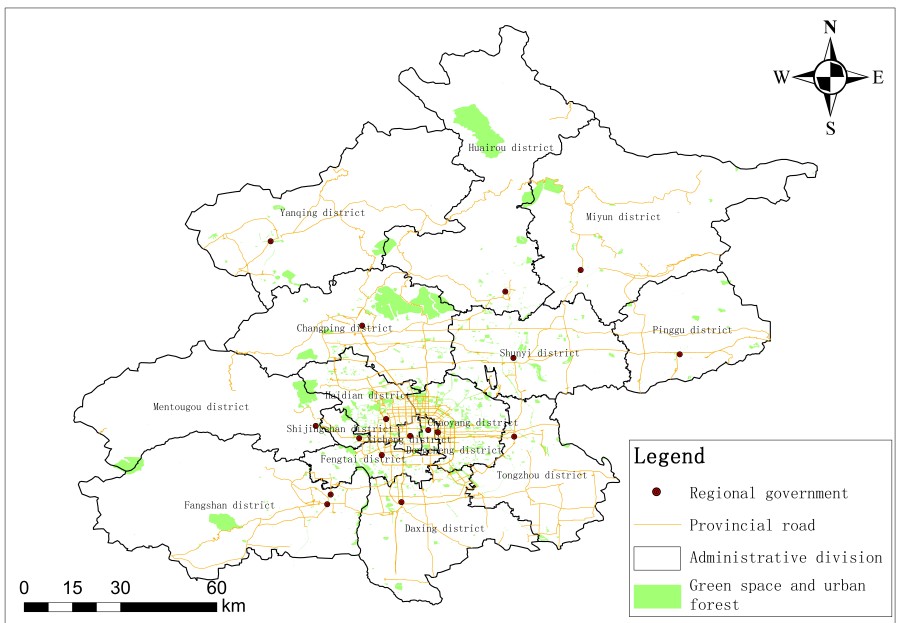

**Figure 2.** Green spaces and urban forests distribution in Beijing in 2015.

*1.2. Literature Review*

Urban green space in China is divided into five parts. These include (1) park green space, (2) protection green space, (3) green space in the square, (4) attached green space, and (5) regional green space [18]. Urban forest is defined as forest or trees planted and managed in or near an urban area. It is mainly composed of natural or transformed forest vegetation. It is also the general term for urban green space where the forest is its main part, which includes vegetation along urban streets and in urban parks, woodlots, abandoned sites, and residential areas, which can constitute an important percent of a nation's tree cover [19].

There are a number of ecosystem services (hereafter referred to as ESs), and they can be classified by different types: provisioning services, regulating services, habitat (or supporting) services, and cultural services. We identified 18 kinds of ESs relevant to urban settings based on the literature. In particular, the Millennium Ecosystem Assessment Synthesis Report (2005), IPBES Conceptual Framework (2015), Mather et al. (2011), Aylor (1972), and Haase (2014) were referenced [20–24].

Urban forests play a very significant role in urban ecosystems through providing a variety of important ESs for people, such as carbon storage and carbon sequestration [25,26], air quality improvement [25–27], water storage [28], recreation and aesthetic services [29,30], microclimate regulation [30], rainwater retention [30], health and psychological services [25,29,30], biodiversity conservation [30], education and sites for scientific research [30], energy conservation [27], wildlife habitats [30], and noise reduction [23,25,29].

The fresh water accessible to the people in Beijing mainly includes surface water, groundwater, and transit water, which is function of the total runoff [31]. Biao [32] reported that the amount of surface runoff in broadleaved forest (such as Q. liaotungensis) was 31.31 m$^3$/ha, and the broadleaved forest was the largest contributor to the service of fresh water provision in Beijing.

A law was passed in the People's Republic of China concerning the regulation of environmental pollution. Stipulated in the law, noise pollution cannot exceed 45 dB. Aylor [23] reported that the foliage of trees reduces noise and absorbs noise pollution. In all of the vegetation belts examined, shrubs were the most effective at reducing noise, owing to

scattering from their dense foliage and branches [33]. According to Ba [34], the tree crown diameter is positively correlated to noise reduction. If the tree crown diameter is small, the ES of noise reduction is relatively low; on the contrary, if the tree crown diameter is large, the ES of noise reduction is relatively high. The average tree crown diameter of a single plant is 0.679 m [34]. Among them, 0.34 is half of the average (0.679), and 1.02 is 1.5 times the average.

He [35] mentioned that stand density is positively correlated to soil conservation. If stand density is small, the ES of soil conservation is relatively low; on the contrary, if the stand density is large, the ES of soil conservation is relatively high. The average stand density is 1400 trees/ha [35]. Among, 700 is half of the average (1400), and 2100 is 1.5 times the average.

Miller [36] mentioned that it is necessary to conserve surface soil as it supports the increase of soil pore spaces, which also contributes to the effective prevention of soil loss and landslide. Zhao and Ouyang [37] reported that the coverage of low-lying vegetation contributes to the prevention of soil loss and landslide by maintaining pore spaces within the soil. The coverage of low-lying vegetation is positively correlated to the prevention of soil loss and landslide.

Garden plants have a significant ability to remove air particulate matters (PM) and can effectively improve the urban environmental quality [38]. Nowak [15] reported that urban trees' contribution to air pollution reduction, the pollution removal of pollutants, is affected by the amount of tree crown coverage in the city. That is, if the tree crown coverage is small, the air quality regulation is relatively low. On the contrary, if the tree crown coverage is large, the air quality regulation is relatively high. Li [39] reported that a 1 $hm^2$ broadleaved forest can absorb 1000 kg $CO_2$ and release 730 kg $O_2$. According to Wang [40], greenspaces can absorb 1.767 ton/ha/day of $CO_2$ and release 1.23 ton/ha/day of $O_2$, and arbor forest took up the largest proportion. The capacity of tall trees to mitigate $CO_2$ is larger than that of shrub. The higher the ratio of tall trees, the larger the carbon sequestration of the forest.

Alvey [19] reported that urban forests play a significant role in maintaining wildlife habitats. Piao [41] grouped the trees into two levels: fruit trees and non-fruit trees, based on the assumption that more fruit trees can attract more birds and animals by providing fruits. While taking into consideration species diversity and wildlife habitat, the number of plant species can be used as an indicator for biodiversity of urban forests; this can be classified into poor, middle, or rich [42]. There are 393 plants species, which belonged to 251 genera and 90 families in Beijing parks [43]. Among, 197 is half of the average (393 species), and 590 is 1.5 times the average.

According to Piao [41], the aesthetic function of urban trees includes the seasonal dynamics of the leaves and the colors of the trees' flowers. We followed the classification of Li [44], who used the change of leaves' color as the criteria. Urban forests are classified into evergreen forest and non-evergreen forest according to whether there is a seasonal color change of leaf [45]. Trails are the pathways where visitors spend most of their time in urban forests. Koo [42] reported that trails are regarded as an important part of the forest recreation infrastructure in Korean society and culture. The levels of trails' density were expressed by the amount of time spent by a visitor walking in the urban forests. These levels were set according to the average time (1 h per day) Beijing citizens spend time in urban forests [46]. The larger the trail density of the forest, the more people can enjoy the forest ES of recreation and therapy.

Municipality tax is a special tax for the management of urban forests every year. Shi [47] set the annual tax per household attribute levels at 0, 5, 10, 20, 50, and 100 RMB in order to calculate the value for enhancing and maintaining the public function/utility of Wenjiang forests. The price that the surveyed citizens are willing to pay for protection of old and famous trees in Beijing was 10–20 RMB/person/year [48]. Zhang and Qi [49] reported that Beijing citizens were willing to pay 50–100 RMB/household/year for governance haze. Considering the characteristics, scope, and targets of previous studies, in this paper, we set

the attribute levels for the annual municipality tax to be 25, 50, and 100 RMB/household (3.77, 7.54, and 15.08 USD/household), as these prices are located between the three price ranges (1 RMB = 6.63 USD).

Trade-off occurs when supplying one ecosystem service (hereafter referred to as ES) is reduced in order to increase the use of another ES. In other words, trade-off means that as the function of an ES increases, the function of a specific ES decreases. Synergy means that as the function of an ES increases, the function of a specific ES increases. It may vary depending on the characteristics of the target site. Even among the services of the same species, trade-off effects may happen in specific areas, and synergy may occur in other areas. According to the results of previous research, although the level and amount are different, it can be seen that trade-off effect and synergy occur simultaneously in all major classification services [24,50–63]. Trade-off and synergy among several different ESs are closely related to the stability of the ecosystem [64].

### 1.3. Rationale for the Study

Zheng [16] indicated that Beijing's urban forests have several problems. These include the presently poor stand quality, along with a low forest coverage rate per capita and relatively backward management level. Urban forest is different from traditional forest in policy-making, as it is expressed through the public's participation in negotiations. That is to say, urban forestry is closely associated with the public [65]. Based on these aspects of Beijing's background, we chose Beijing urban green space as our study area. Due to the low urban green space per capita in Beijing, more attention should be paid to the designing and managing of urban forests.

The period of the Thirteenth Five-Year Plan (2016–2020) is an important time for building a prosperous society in a comprehensive way and constructing an ecological civilization and beautiful China. It is also the crucial period during which Beijing is supposed to achieve the strategic goal of becoming an international first-class, harmonious, sustainable, and livable city. Urban tree cover provides a range of ecosystem services [66]. In many countries, developing urban forest is a significant strategy for a city's sustainable development [67]. Urban forest management has become an important issue in urban management policy [2].

In order to effectively design and manage urban forests, more attention should be paid to effectively promoting resident's satisfaction regarding urban forests. It is important to research Beijing resident's preferences regarding urban forests in terms of different types of ESs. However, little is known about Beijing resident's preferences regarding various ESs in urban forests from their own perspective. This paper aims to provide policy decision-makers with key information for the development and maintenance of urban forests. To elicit the relative preferences to the ecosystem service of Beijing residents, we employed the choice experiment [68] mothed. The results will make recommendations for efficient urban forest management.

This study is designed with the following three objectives: (1) to investigate urban resident's preferences and options regarding various ESs of urban forests; (2) to know how much Beijing residents are willing to pay for various ESs of urban forests; and (3) to analyze the preferences of Beijing residents based on their sociodemographic characteristics regarding various ESs in urban forests. Understanding Beijing resident's perspectives of urban forest ESs will provide policy options for meeting the public's needs, improving the distribution and supply of ESs, and minimizing conflicts between policy makers and ESs. It would also help Beijing work toward its strategic goal of becoming an international first-class, harmonious, sustainable, and livable city. There are four questions: (1) What is the most prevalent preference of Beijing residents regarding various ESs in urban parks? (2) How much are Beijing residents willing to pay for various ESs in urban parks? (3) Do Beijing residents living in an apartment owned by themselves differ from apartment non-owning dwellers in terms of preference to the attribute level for each ES? (4) Are

Beijing residents with high income more willing to pay a municipality tax than low-income residents?

According to the questions, there are three hypotheses: (1) Beijing residents regard air quality improvement and water provision as the most important ESs. (2) Apartment owners are willing to pay more municipality tax for forest ESs than apartment non-owning dwellers. (3) Residents with high income are more willing to pay a municipality tax than low-income residents. Following this paper, those questions will be answered.

## 2. Materials and Methods

### 2.1. The Delphi Analysis

The Delphi method is a popular tool for modern foresight in many countries [69]. It was developed by employees of the Rand Corporation in the 1950s. Since that time, it has become a widely used evaluation research technique. It is seen as a process of obtaining the most reliable opinions and consensus of a group of experts through a series of questionnaires interspersed with feedback [70,71].

The concept and measurement scheme of urban forest ESs were explained to the panel of experts online with help of a questionnaire to investigate their perspectives on the relative importance of the 18 types of ESs chosen based on the literature review [20–24], including food supply [72], raw materials provision [73], fresh water provision [74], medicinal resources provision [75], local climate and air quality regulation [76], carbon sequestration and storage [77], moderation of extreme events [78], wastewater treatment [79], erosion prevention and the maintenance of soil fertility [80], pollination [81], biological control [82], noise reduction [23,25,29], habitats for species [83], maintenance of genetic diversity [84], recreation and mental and physical health [54], tourism [85], aesthetic appreciation and inspiration for culture, art and design [22], spiritual experience, and sense of place [22].

The recruited 30 experts were specialists in forestry and landscape architecture. Fifteen of them are public officials employed by the Beijing Gardening and Greening Bureau, and 15 experts are scholars engaged in forestry and ecology research. They were contacted via email and Wenjuanxing, which is a professional online questionnaire survey, evaluation, and voting platform in China. In total, 30 effective responses were received.

There are two distinct relationships between ecosystem services, trade-off and synergy [63], that should be considered when experts answering the survey questions. 1st, 2nd, and 3rd represent the importance degree of urban forest ecosystem services for Beijing residents, respectively, in which 1st is important, 2nd is of medium importance, and 3rd is unimportant.

According to the "Importance Value = #A $*$ 3 + #B $*$ 2 + #C $*$ 1", the first experts' online survey result is as follows (Table 1).

According to the above result, the important rank is as follows (Table 2).

The first conclusive experts' responses were again sent to the same 30 experts. They were asked if they would change their responses after viewing their initial results. The second round of results concluded with 30 effective responses (Table 3).

Comparing the results of the first and second surveys, the importance value rank of local climate and air quality regulation ES did not change, which is the most important among the 18 types ESs in the two surveys. In addition, the 5 lowest-ranking ES types (wastewater treatment, pollination, food supply, medicinal resources provision, and raw materials provision) did not change the rank of ESs importance value in the two surveys, which were removed from consideration due to their relatively low importance value, left 13 types of ESs.

**Table 1.** Importance value of ecosystem services (ESs) in the first experts' online survey.

| Ecosystem Services | Rank the Importance | | | Importance Value [1] |
|---|---|---|---|---|
| | A = 1st [a] | B = 2nd [a] | C = 3rd [a] | |
| 1. Food supply | 11 | 7 | 12 | 59 |
| 2. Raw materials provision | 5 | 6 | 19 | 46 |
| 3. Fresh water provision | 22 | 8 | 0 | 82 |
| 4. Medicinal resources provision | 8 | 8 | 13 | 56 |
| 5. Local climate and air quality regulation | 30 | 0 | 0 | 90 |
| 6. Carbon sequestration and storage | 18 | 12 | 0 | 78 |
| 7. Moderation of extreme events | 22 | 8 | 0 | 82 |
| 8. Wastewater treatment | 12 | 16 | 2 | 70 |
| 9. Erosion prevention and maintenance of soil fertility | 21 | 8 | 1 | 80 |
| 10. Pollination | 8 | 19 | 3 | 65 |
| 11. Biological control | 19 | 10 | 1 | 78 |
| 12. Noise reduction | 21 | 9 | 0 | 81 |
| 13. Habitats for species | 20 | 9 | 1 | 79 |
| 14. Maintenance of genetic diversity | 16 | 12 | 2 | 74 |
| 15. Recreation and mental and physical health | 24 | 6 | 0 | 84 |
| 16. Tourism | 18 | 12 | 0 | 78 |
| 17. Aesthetic appreciation and inspiration for culture, art and design | 22 | 7 | 1 | 81 |
| 18. Spiritual experience and sense of place | 23 | 7 | 0 | 83 |

[a] The number of experts who thought the importance level of urban forest ESs for Beijing residents. [1] Importance Value = #A * 3 + #B * 2 + #C * 1. A = the number of experts who thought this specific ES of urban forest is important; B = the number of experts who thought this specific ES of urban forest is medium important; C = the number of experts who thought this specific ES of urban forest is unimportant.

**Table 2.** Ranking the importance value of ESs in the first experts' online survey.

| Rank | ESs | Importance Value [1] |
|---|---|---|
| 1 | Local climate and air quality regulation | 90 |
| 2 | Recreation and mental and physical health | 84 |
| 3 | Spiritual experience and sense of place | 83 |
| 4 | Moderation of extreme events | 82 |
| 5 | Fresh water provision | 82 |
| 6 | Aesthetic appreciation and inspiration for culture, art, and design | 81 |
| 7 | Noise reduction | 81 |
| 8 | Erosion prevention and maintenance of soil fertility | 80 |
| 9 | Habitats for species | 79 |
| 10 | Carbon sequestration and storage | 78 |
| 11 | Tourism | 78 |
| 12 | Biological control | 78 |
| 13 | Maintenance of genetic diversity | 74 |
| 14 | Wastewater treatment | 70 |
| 15 | Pollination | 65 |
| 16 | Food supply | 59 |
| 17 | Medicinal resources provision | 56 |
| 18 | Raw materials provision | 46 |

[1] Importance Value = #A * 3 + #B * 2 + #C * 1. A = the number of experts who thought this specific ES of urban forest is important; B = the number of experts who thought this specific ES of urban forest is medium important; C = the number of experts who thought this specific ES of urban forest is unimportant.

**Table 3.** Ranking the importance value of ESs in the second experts' online survey.

| Rank | ESs | Importance Value [1] |
|---|---|---|
| 1 | Local climate and air quality regulation | 90 |
| 2 | Carbon sequestration and storage | 86 |
| 3 | Recreation and mental and physical health | 74 |
| 4 | Fresh water provision | 72 |
| 5 | Noise reduction | 72 |
| 6 | Moderation of extreme events | 69 |
| 7 | Spiritual experience and sense of place | 69 |
| 8 | Erosion prevention and maintenance of soil fertility | 67 |
| 9 | Aesthetic appreciation and inspiration for culture, art and design | 65 |
| 10 | Maintenance of genetic diversity | 62 |
| 11 | Tourism | 58 |
| 12 | Habitats for species | 57 |
| 13 | Biological control | 52 |
| 14 | Wastewater treatment | 42 |
| 15 | Pollination | 38 |
| 16 | Food supply | 36 |
| 17 | Medicinal resources provision | 35 |
| 18 | Raw materials provision | 33 |

[1] Importance Value = #A ∗ 3 + #B ∗ 2 + #C ∗ 1. A = the number of experts who thought this specific ES of urban forest is important; B = the number of experts who thought this specific ES of urban forest is of medium importance; C = the number of experts who thought this specific ES of urban forest is unimportant.

### 2.2. Regrouping 13 ESs into 6 Groups

In third survey, 30 experts were surveyed regarding regrouping 13 types of ESs. This method involved conducting a survey about the respondents' opinions regarding the relationship between two different ecosystem services. In this case, 1–5 represented the relationship between 2 types of ecosystem services (1—very closely related; 2—closely related; 3—neutral; 4—not closely related; 5—unrelated).

In this case, smaller numbers meant a closer relationship. In addition, it is more likely that the ESs sharing close relationships could be combined to form a single group. Then, the closely related ESs were regrouped based on mutual relationships.

According to experts' third survey, 13 types of ESs were regrouped into 6 groups, which are fresh water provision, noise reduction, water/soil conservation (including moderation of extreme events), climate and air quality regulation (including carbon sequestration and storage), biodiversity conservation (including habitats for species, biological control, maintenance of genetic diversity), and cultural service (including recreation, tourism, aesthetic appreciation, spiritual experience).

### 2.3. Choice Experiment (CE)

The literature review indicated that there are some urban forest ES attributes and attribute levels. The fourth survey focused on 30 experts and helped define the correct ES attributes and attribute levels. It resulted in 30 effective responses (Table 4).

**Table 4.** The attribute, indicator, and attribute level for ES.

| Ecosystem Service | Attribute | Indicator | Attribute Level |
|---|---|---|---|
| 1. Fresh water provision | Fresh water provision (FWP) | Proportion of broadleaf trees | The ES of fresh water provision is low if it only has softwood. |
| | | | The ES of fresh water provision is high if it only has hardwood. |
| 2. Noise reduction | Noise reduction (NR) | Floral composition | The ES of noise reduction is low if only trees are present. |
| | | | The ES of noise reduction is high if trees and shrubs are present. |
| 3. Water/soil conservation (including moderation of extreme events) | Moderation of extreme events (MEE) | Coverage of low-lying vegetation | The prevention of soil loss and landslide is low if the coverage of low-lying vegetation is below 30%. |
| | | | The prevention of soil loss and landslide is high if the coverage of low-lying vegetation is above 70%. |
| 4. Climate and air quality regulation (including climate change mitigation) | Air quality regulation (AQR) | Tree crown coverage | The ES of air quality regulation is low if tree crown coverage is below 25%. |
| | | | The ES of air quality regulation is middle if tree crown coverage is 25–75%. |
| | | | The ES of air quality regulation is high if tree crown coverage is above 75%. |
| 5. Biodiversity conservation | Species diversity and wildlife habitat (SDWH) | The number of plant species/km$^2$ | The species diversity and wildlife habitat is low if the forest is composed of a single species of tree. |
| | | | The species diversity and wildlife habitat is rich if the forest is composed of multiple species of tree and shrubs. |
| 6. Cultural service (including recreation, tourism, aesthetic appreciation and spiritual experience) | Recreation and spiritual experience (RSE) | Density of trails | The ES of recreation and therapy service is low if the density of trails is low. |
| | | | The ES of recreation and therapy service is medium if the density of trails is medium. |
| | | | The ES of recreation and therapy service is high if the density of trails is high. |
| 7. Willingness-to-pay | Municipality tax (MT) | Level of payment | 25 RMB ($3.53)/per year |
| | | | 50 RMB ($7.06)/per year |
| | | | 75 RMB ($10.59)/per year |

*2.4. Experimental Design*

A total of 576 combinations can be created with 4 attributes with 2 levels, 2 attributes with 3 levels, and 1 attribute with 4 levels of urban forest ecosystem services. It is unfeasible to develop a questionnaire containing all of these combinations, so the number of alternatives was reduced by using an SPSS orthogonal design procedure. The SPSS procedure produced 16 alternatives (Table 5). The 16 alternatives were also randomly divided into 8 different versions, each with 2 choice sets. This was to further simplify the survey format.

A choice set consisted of 2 management scenario profiles and an option to select neither scenario. Each interviewer was asked four times to choose a set. Table 6 shows an example of a questionnaire with this choice set.

The questionnaire was comprised of 4 parts. The first part was attitudinal questions. These included the frequency of visits, visiting motives, the usages and the perception of urban forests, and the likeability of Beijing. The second part included descriptions of the attributes of the choice experiment. This consisted of questions regarding the importance of 6 types of urban forest ecosystem services. The third part was a choice experiment—4 choice sets (questions), each with 2 alternatives and 1 optional alternative. The forth part considered socioeconomic data. This focused on questions about age, gender, marriage, number of children under 20 years old, education level, employment, if the job is environment or forest-related, income level, if the respondent has a history of living in a rural area, the number of years spent living in the countryside, and if they have an apartment in Beijing.

### 2.5. Data Collection

Beijing has a total population of 21.729 million as of 2016. The urban population of Beijing was 18.796 million in 2016 [86]. With a margin of error set to 4–5% at a 95% confidence level, a sample size consisting of 384 to 600 people was deemed to be the most appropriate. In total, 560 interviews with questionnaires with 8 versions were complemented in the 16 districts of Beijing. There were 56 respondents who expressed a very poor or poor ability in understanding the information provided in the questionnaire. In addition, 21 questionnaires were incomplete. As a result, 77 invalid questionnaires were removed from the analysis, leaving a total of 483 effective questionnaires for analysis. Table 7 shows that the valid sample can almost confirm the principle of the population density ratio. It also is an accurate indication as to the entire population of Beijing.

In total, 5 surveys were conducted. Table 8 shows response-related statistics. From 27 October to 4 November, the final survey (field survey) was conducted. This examined 560 Beijing residents who were aged 20 or more than 20 years old, had visited Beijing urban parks during 2016, and had lived in Beijing for 1 year or more than 1 year.

### 2.6. Model Estimation (Conditional Logit Model)

Equation (1) is based on McFadden's Random utility model [87]

$$U_{nj} = V_{nj} + e_{nj} \tag{1}$$

when the subject ($n$) chooses the alternative ($j$), Indirect utility function $U_{nj}$ is formed with the fixed part $V_{nj}$ and probability part $e_{nj}$. Following the demand characteristics theory [88], the fixed $V_{nj}$ is formed with ($n$) number of attributive vectors.

According to the demand characteristics theory, Equation (2) is comprised of the linear of sum of the number of $n$ attribute's vector.

$$V_{ni} = \sum_{k=1}^{m} \beta_k X_i \tag{2}$$

$X_i$, Equation (3) shows the probability for ($n$) number of respondents to select $j$ instead of $i$ based on the discrete choice model [89]. In other words, the utility for choosing $j$ is bigger than the utility for choosing $i$. The discrete choice model is based on the utility maximization theory, in which all individual attempt to make decisions by seeking the maximization of their utilities.

$$P_{nj} = \Pr(V_{nj} + e_{nj} > V + e) \tag{3}$$

**Table 5.** Card List of 16 alternatives of urban forest ecosystem services.

| | FWP | NR | MEE | AQR | SDWH | RSE | MT [1] |
|---|---|---|---|---|---|---|---|
| 1 | only softwood | trees and shrubs | low-lying vegetation below 30% | tree crown coverage 25–75% | plant species ranges 197–590 | low density of trails | 50 |
| 2 | only softwood | trees and shrubs | low-lying vegetation above 70% | tree crown coverage below 25% | plant species ranges 197–590 | high density of trails | 200 |
| 3 | only hardwood | trees and shrubs | low-lying vegetation above 70% | tree crown coverage below 25% | plant species ranges 197–590 | medium density of trails | 50 |
| 4 | only softwood | trees and shrubs | low-lying vegetation above 70% | tree crown coverage above 75% | plant species ranges above 590 | low density of trails | 100 |
| 5 | only hardwood | only trees | low-lying vegetation below 30% | tree crown coverage below 25% | plant species ranges 197–590 | low density of trails | 25 |
| 6 | only hardwood | trees and shrubs | low-lying vegetation below 30% | tree crown coverage above 75% | plant species ranges 197–590 | low density of trails | 200 |
| 7 | only softwood | trees and shrubs | low-lying vegetation below 30% | tree crown coverage below 25% | plant species ranges above 590 | high density of trails | 25 |
| 8 | only hardwood | only trees | low-lying vegetation above 70% | tree crown coverage 25–75% | plant species ranges 197–590 | high density of trails | 100 |
| 9 | only hardwood | trees and shrubs | low-lying vegetation below 30% | tree crown coverage below 25% | plant species ranges above 590 | medium density of trails | 100 |
| 10 | only softwood | only trees | low-lying vegetation below 30% | tree crown coverage below 25% | plant species ranges 197–590 | low density of trails | 100 |
| 11 | only softwood | only trees | low-lying vegetation below 30% | tree crown coverage 25–75% | plant species ranges above 590 | medium density of trails | 200 |
| 12 | only softwood | only trees | low-lying vegetation above 70% | tree crown coverage above 75% | plant species ranges 197–590 | medium density of trails | 25 |
| 13 | only hardwood | trees and shrubs | low-lying vegetation above 70% | tree crown coverage 25–75% | plant species ranges above 590 | low density of trails | 25 |
| 14 | only softwood | only trees | low-lying vegetation above 70% | tree crown coverage below 25% | plant species ranges above 590 | low density of trails | 50 |
| 15 | only hardwood | only trees | low-lying vegetation above 70% | tree crown coverage below 25% | plant species ranges above 590 | low density of trails | 200 |
| 16 | only hardwood | only trees | low-lying vegetation below 30% | tree crown coverage above 75% | plant species ranges above 590 | high density of trails | 50 |

[1] Unit: RMB.

**Table 6.** Example of questionnaire with the choice set.

**Which of the Following Urban Forest Ecosystem Services Do You Favor? Option A and Option B Would Entail a Cost to Your Household. No Payment Would Be Required for "Neither A nor B" Option.**

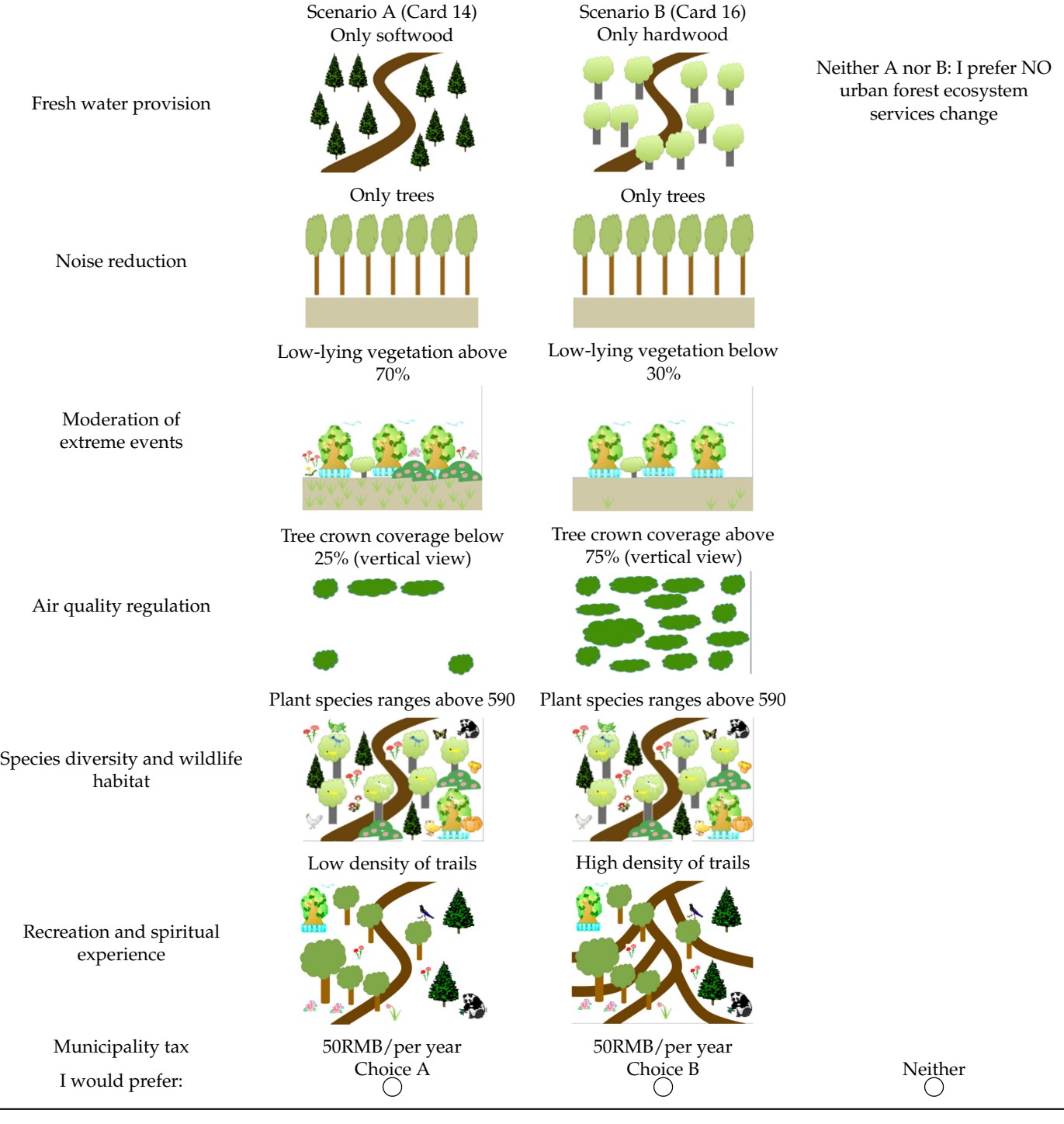

| | Scenario A (Card 14) Only softwood | Scenario B (Card 16) Only hardwood | Neither A nor B: I prefer NO urban forest ecosystem services change |
|---|---|---|---|
| Fresh water provision | | | |
| Noise reduction | Only trees | Only trees | |
| Moderation of extreme events | Low-lying vegetation above 70% | Low-lying vegetation below 30% | |
| Air quality regulation | Tree crown coverage below 25% (vertical view) | Tree crown coverage above 75% (vertical view) | |
| Species diversity and wildlife habitat | Plant species ranges above 590 | Plant species ranges above 590 | |
| Recreation and spiritual experience | Low density of trails | High density of trails | |
| Municipality tax | 50RMB/per year | 50RMB/per year | |
| I would prefer: | Choice A ◯ | Choice B ◯ | Neither ◯ |

**Table 7.** Population of region sample.

| | Region | Census Register Household [1] | Resident Population [1] | Sample (In Total: 560) | Valid Sample (In Total: 483) |
|---|---|---|---|---|---|
| 1 | Dongcheng district | 34.6 | 87.8 | 36 | 32 |
| 2 | Xicheng district | 48.5 | 125.9 | 51 | 42 |
| 3 | Chaoyang district | 81.0 | 385.6 | 86 | 68 |
| 4 | Fengtai district | 47.0 | 225.5 | 50 | 39 |
| 5 | Shijingshan district | 14.7 | 63.4 | 16 | 14 |
| 6 | Haidian district | 72.4 | 359.3 | 77 | 77 |
| 7 | Shunyi district | 27.0 | 107.5 | 29 | 24 |
| 8 | Tongzhou district | 35.4 | 142.8 | 37 | 31 |
| 9 | Daxing district | 26.9 | 169.4 | 28 | 26 |
| 10 | Fangshan district | 37.8 | 109.6 | 40 | 38 |
| 11 | Mentougou district | 12.1 | 31.1 | 13 | 13 |
| 12 | Changping district | 26.6 | 201.0 | 28 | 26 |
| 13 | Pinggu district | 17.0 | 43.7 | 18 | 13 |
| 14 | Miyun district | 20.7 | 48.3 | 22 | 14 |
| 15 | Huairou district | 13.6 | 39.3 | 14 | 13 |
| 16 | Yanqing district | 14.0 | 32.7 | 15 | 13 |

[1] Unit: ten thousand, census data.

**Table 8.** Response statistics.

| Survey | | Case Number (People) | Effective Response Number | Response Proportion (%) |
|---|---|---|---|---|
| The first and second online experts surveys (Delphi analysis) | First | 30 | 30 | 100 |
| | Second | 30 | 30 | 100 |
| The third online experts survey for regrouping ESs | | 30 | 22 | 73.3 |
| The fourth online experts survey for defining the urban forest ES attributes and attribute levels | | 30 | 30 | 100 |
| The final field survey in Beijing | | 560 | 483 | 86 |

In the conditional logit model [90], the parameter value was estimated assuming the probability part based on Gumble distribution/Extreme value type I distribution shown in Equation (1).

The probability distribution is shown in Equation (4).

$$P_{nj} = \frac{e^{\beta' Xni}}{\sum_i e^{\beta' Xni}} \tag{4}$$

For the data collected from survey, there 7 independent variables that may influence the probability of the resident's choice over forest ESs. The empirical model is specified as below.

$X_{ni}$ = 1, 2, 3, 4, 5, 6, 7;

$X_1$ = Fresh water provision;

$X_2$ = Noise reduction;

$X_3$ = Moderation of extreme events;

$X_4$ = Air quality regulation;

$X_5$ = Species diversity and wildlife habitat;

$X_6$ = Recreation and spiritual experience;

$X_7$ = Municipality tax.

**3. Results**

*3.1. Sociodemographic Characteristics of Respondents*

Here were the results of the surveys. Table 9 showed the respondents' sociodemographic characteristics, and the details of each part are provided in Table 10.

**Table 9.** Description of the sample.

|  | N | Minimum | Maximum | Mean | Std. Deviation |
|---|---|---|---|---|---|
| Gender [1] | 483 | 0.00 | 1.00 | 0.48 | 0.50 |
| Age | 483 | 20.00 | 84.00 | 40.52 | 14.71 |
| Education level [2] | 483 | 1.00 | 5.00 | 3.68 | 0.90 |
| Monthly household income [3] | 483 | 1.00 | 8.00 | 3.35 | 1.29 |
| Years lived in countryside [4] | 483 | 0.00 | 3.00 | 1.39 | 1.33 |
| Apartment owner [5] | 483 | 0.00 | 1.00 | 0.43 | 0.50 |
| Relative environmental work [6] | 483 | 0.00 | 1.00 | 0.12 | 0.33 |

[1] Male = 0, Female = 1; [2] Elementary school graduate = 1, Middle school graduate = 2, High school graduate = 3, University degree holder = 4, Graduate school student or graduate degree holder = 5; [3] Below 5000 RMB = 1; 5000–9999 RMB = 2, 10,000–14,999 RMB = 3, 15,000–19,999 RMB = 4, 20,000–24,999 RMB = 5, 25,000–29,999 RMB = 6, 30,000–34,999 RMB = 7, Over 35,000 RMB = 8; [4] 0 year = 0, Below 1 year (remove 0) = 1, 1–3 year(s) = 2, Above 3 years = 3; [5] Non-apartment owner = 0, Apartment owner = 1; [6] Irrelative environmental work = 0, Relative environmental work = 1.

**Table 10.** Descriptive statistics of respondents' characteristics.

| Characteristics | | Sample N = 483 | Proportion of Population (%) (Census) |
|---|---|---|---|
| Gender | Male | 51.6 | 51.2 |
| | Female | 48.4 | 48.8 |
| Age | 20–29 | 29.4 | 20.6 |
| | 30–39 | 25.0 | 19.6 |
| | 40–49 | 17.3 | 16.0 |
| | 50–59 | 14.3 | 14.8 |
| | 60 and above | 13.9 | 15.9 |
| Monthly household income | Below 5000 (RMB) | 5.0 | |
| | 5000–9999 (RMB) | 23.2 | |
| | 10,000–14,999 (RMB) | 28.4 | |
| | 15,000–19,999 (RMB) | 24.8 | |
| | 20,000–24,999 (RMB) | 14.9 | |
| | 25,000–29,999 (RMB) | 1.9 | |
| | 30,000–34,999 (RMB) | 1.2 | |
| | Over 35,000 (RMB) | 0.6 | |
| Apartment owner | Yes | 56.5 | |
| | No | 43.5 | |
| Marriage | Married | 69.2 | |
| | Single | 30.8 | |
| Number of children (under 20 years old) | None | 65.2 | |
| | 1 | 31.7 | |
| | 2 | 3.1 | |
| | 3 and above | 0 | |
| Education level | Elementary school graduate | 0.8 | |
| | Middle school graduate | 7.9 | |
| | High school graduate | 32.7 | |
| | University degree holder | 39.3 | |
| | Graduate school student or graduate degree holder | 19.3 | |
| Work related to environment or forest | Yes | 87.6 | |
| | No | 12.4 | |

The results related to statistics for respondents' sociodemographic characteristics revealed specific information regarding the respondents. Their gender mean was 0.48, which showed that the gender ratio was almost balanced. Their ages ranged from 20 to 84 years old. Their education level mean was 3.68, which showed that it was centered on a level between a high school diploma holder and someone with a university degree. Their monthly household income mean was 3.35, which indicated that the respondents'

monthly household income was almost 14,316 RMB. The mean for years lived in the countryside was 1.39, which indicated that most of them had lived in the countryside for 1 year. The mean for apartment owners was 0.43, which showed that 43.5% of respondents did not own one located in Beijing. The respondents' jobs were seldom related to the environment or forest.

The results of the statistics for respondents' characteristics in Table 10 showed that 69.2% of the respondents were married and 30.8% of the respondents were single, 65.2% of the respondents did not have children under 20 years old, and 31.7% of the respondents had one child under 20 years old.

### 3.2. Results of Model Estimation

Table 11 used the STATA conditional logit model to estimate the empirical model results of the base model (without considering socioeconomic interactions), which were specified in Equation (3). Dummy coding was used to code all of the qualitative variables. The coefficient was the estimated parameter, which was used to calculate the utility provided by the change in the given attribute. The coefficient indicated the direction of movement of the utility derived from an increase in the level of the attribute. A larger coefficient meant that it would have a stronger effect on the probability of residents preferring an ES choice. That is to say, a positive coefficient indicated that an increase in the attribute level would increase the utility provided. On the contrary, a negative coefficient showed that an increase in the attribute level would decrease the utility provided with all other conditions remaining constant. The standard error (SE) was used to measure the sampling error. The standard error was smaller, and the sample statistic was closer to the value of the population parameter, which meant that the sample was more representative to the population, and the sample statistic was more reliable to infer the population parameter. The P-value indicated the risk level at which the null hypothesis can be rejected. The R2 showed how much of the choice behavior the model could explain. Pseudo R2 helped to understand whether R2 made sense. An example of this was to suppose that the covariates in the current model did not actually provide any predictive information regarding the outcome.

**Table 11.** Estimated conditional logit model.

| Choice | | Coef. | Std.Err. | $p > |z|$ |
|---|---|---|---|---|
| FWP (base = low) | high | 0.863 | 0.088 | 0.000 |
| NR (base = low) | high | 0.183 * | 0.098 | 0.062 |
| MEE (base = low) | high | −0.165 | 0.079 | 0.037 |
| AQR (base = low) | medium | 0.564 *** | 0.097 | 0.000 |
| | high | 1.743 *** | 0.113 | 0.000 |
| SDWH (base = low) | high | −0.122 * | 0.087 | 0.164 |
| RSE (base = low) | medium | 0.619 ** | 0.104 | 0.000 |
| | high | −0.200 | 0.100 | 0.046 |
| No. of Observation | | 3864 | | |
| Pseudo R2 | | 0.274 | | |
| Log likelihood | | −931.598 | | |

Significant levels: * 10%, ** 5%, *** 1%.

The conditional logit model did not account for preference heterogeneity, which meant individuals did not express their own identical preferences when choosing alternatives between choice cards. Namely, the conditional logit model in Table 11 is based on the assumption that residents have definite preferences when choosing the choices in the survey.

Table 12 showed that Beijing residents were willing to pay for municipality tax for different ESs. The order of willingness-to-pay (WTP) for various ESs from large to small: air quality regulation from low to high, fresh water provision from low to high, recreation and spiritual experience from low to medium, air quality regulation from low to

medium, noise reduction from low to high, species diversity and wildlife habitat from low to high, moderation of extreme events from low to high, and recreation and spiritual experience from low to high. Beijing residents were willing to pay for expanding urban forest ecosystems in order to improve air quality, up to 264.065 Chinese RMB per year, which is equivalent to 1.84% of the average monthly income household of Chinese citizens annually for improved air quality from a low level to a high level.

**Table 12.** Calculated willingness-to-pay (WTP) for municipality tax for various ESs.

| Attributes | | Mean WTP | 95% CI | |
|---|---|---|---|---|
| | | | Minimum | Maximum |
| FWP (base = low) | high | 130.725 | 97.397 | 164.053 |
| NR (base = low) | high | 27.765 | −0.987 | 56.517 |
| MEE (base = low) | high | −25.026 | −47.566 | −2.486 |
| AQR (base = low) | medium | 85.519 | 52.996 | 118.042 |
| | high | 264.065 | 207.932 | 320.197 |
| SDWH (base = low) | high | −18.432 | −43.644 | 6.781 |
| RSE (base = low) | medium | 93.819 | 58.807 | 128.831 |
| | high | −30.325 | −59.050 | −1.600 |

### 3.3. Results of Different Respondent Groups

#### 3.3.1. Apartment Owners and Non-Apartment Owners

There were 483 respondents included in this analysis. Two models were estimated here, considering Model (1), or respondents who self-identified as apartment owners and Model (2), or respondents who self-identified as non-apartment owners. The level of willingness to pay a municipality tax per year for each attribute was described in Table 13.

**Table 13.** Comparison of WTP values for apartment owners and non-apartment owners.

| Attributes. | | Model (1) Apartment Owner | Model (2) Non-Apartment Owner |
|---|---|---|---|
| | | Coef. | Coef. |
| MT | | 0.0219 | 0.002 |
| FWP (base = low) | high | 1.765 *** | 0.759 ** |
| NR (base = low) | high | 0.913 ** | 0.025 |
| MEE (base = low) | high | −0.344 | −0.397 |
| AQR (base = low) | medium | 1.554 *** | 0.382 * |
| | high | 3.561 *** | 1.612 *** |
| SDWH (base = low) | high | −1.571 | 0.318 * |
| RSE (base = low) | medium | 0.300 * | 0.692 ** |
| | high | −1.046 * | −0.047 |

Significant levels: * 10%, ** 5%, *** 1%.

Apartment owners expressed a high level of willingness to pay for the fresh water provision, air quality regulation, and noise reduction ESs. If apartment owners were highly correlated to income level, this result could also be attributed to higher income levels. Non-apartment owners were more willing to pay for species diversity and wildlife habitat ES in comparison with apartment owners. Regardless of the individual, they were all unwilling to pay a moderate amount for the extreme events ES and a high amount for the recreation and spiritual experience ES. In the case of the recreation and spiritual experience ES, it showed that residents had no particularly high demand for recreation and spiritual experiences, while a middle level of spending on the recreation and spiritual experience ES was enough.

### 3.3.2. Different Monthly Household Income Levels

This section analyzed 483 respondents. Four models were estimated, in which Model (3) to Model (6)'s respondents represented different monthly household income levels (Table 14).

**Table 14.** Comparison of WTP values for different monthly household income levels.

| Attributes | | Model (3) Income below 10,000 RMB | Model (4) Income 10,000–14,999 RMB | Model (5) Income 15,000–19,999 RMB | Model (6) Income Over 19,999 RMB |
|---|---|---|---|---|---|
| | | Coef. | Coef. | Coef. | Coef. |
| Municipality Tax | | −0.001 | 0.007 | 0.013 | 0.027 |
| FWP (base = low) | high | 0.997 * | 0.852 ** | 0.959 ** | 2.270 *** |
| NR (base = low) | high | −0.399 | 0.164 | 0.655 ** | 0.867 ** |
| MEE (base = low) | high | −1.118 | −0.143 | −0.065 | −0.039 |
| AQR (base = low) | medium | −0.315 * | 0.755 ** | 1.300 ** | 2.110 ** |
| | high | 1.645 ** | 1.520 *** | 2.670 *** | 4.457 *** |
| SDWH (base = low) | high | 0.934 * | −0.035 | −0.578 | −2.534 |
| RSE (base = low) | medium | 1.427 * | 0.410 * | −0.131 | 0.358 * |
| | high | 0.173 * | −0.436 | −0.805 | −1.037 |

Significant levels: * 10%, ** 5%, *** 1%.

Residents with high monthly household income were positively related to a willingness to pay the tax for receiving urban forest ESs. The low monthly household income resident's coefficient was minus. It indicated that low monthly household income residents had a negative impact on one's willingness to pay a municipality tax. However, low-income residents were still willing to pay for the fresh water and air quality regulation ESs.

## 4. Discussion

### 4.1. Resident's Preferences of Ecosystem Services of Urban Forests

The result reveals the fact that Beijing citizens are most concerned with the environmental quality and thus are willing to pay the most for regulating services of urban forests, in particular regulating air quality. The second priority of urban forests' ESs regarded by Beijing citizens is the provision of fresh water as revealed in terms of WTP estimates. The first hypothesis of this study can be argued as accepted with this proof. Beijing residents pay little attention to urban forests for the moderation of extreme events such as landslides. A plausible explanation for this is that Beijing citizens are tired of suffering from severe haze and are facing water shortages, while landslides rarely occur in the metropolitan area of Beijing. It was interesting to note that Beijing citizens are willing to pay a moderate amount for recreation and spiritual experiences in association with urban forests but not willing to pay a large amount for cultural services of urban forests. To Beijing citizens, cultural services of urban forests are considered as a kind of luxury that is not essential.

The results of this study are somewhat different from those of previous studies on other cities. Clean water and recreation opportunities are perceived as important ESs for urban residents in Oregon State [91]. Biodiversity is the most significant service for Seoul residents [2,42], aesthetic ES is considered important in Guangzhou [92,93], and recreational ES is perceived as most important in Berlin, Stockholm, Rotterdam, and Salzburg [93]. People in three Alpine regions in Austria and Italy place higher priorities in ESs for meeting basic human needs (e.g., "fresh water", "habitat", "energy", and "food"), followed by regulating and supporting ESs (e.g., "natural hazard regulation", "air quality regulation", "water cycle", and "nutrient cycle") [94]. Differences in the social and physical conditions of cities make the citizens demand different ESs.

The result of this study reveals that apartment owners in Beijing are willing to pay for most of the ESs provided by urban forests in contrast to those residing in other types of

housing. We interpret this result in three ways. One is that non-apartment residents are less likely stay in Beijing for a long time because the living expenditure of Beijing is too high for the poor to survive there. As a result of their uncertain prospects regarding long residency, they may not become very concerned about the development of an ecological environment in Beijing. Another reason for little WTP for urban forests expansion is that the income level of non-apartment residents is relatively low and they cannot afford to pay for non-essential ESs. The fact that the non-apartment owners' WTP is positive indicates that they are interested in improving their living environment. It might be the case that if the burden is not too high, they are also willing to support the city government's policy of expanding urban forests. Another explanation is that Beijing's environmental problems are very serious, and as a result, residents wanted to improve this situation, despite their low income. Beijing residents have a well-developed sense of environmental awareness [95,96]. It means that there is a potential for them to support an environmental protection tax, which can help realize more ecosystem services to be provided by urban forests expanded.

An unexpected result of this study is that Beijing citizens are willing to bear even an increase of the municipality tax for the expansion of urban forests. The positive estimated coefficients for the municipality tax mean that they are willing to pay more municipality tax for an increased supply of ESs needed for improving the quality of the environment, which has been deteriorating. It meant that the ever-deteriorating environmental situation has enforced Beijing residents to be aware of the importance of environmental protection regardless of their financial burden.

*4.2. Policy Implication for Urban Forests*

The Chinese government regards the national development strategy called "ecological civilization" as a very important vision of the nation. In October 2015, with the convening of the fifth Plenary Session of the 18th CPC Central Committee, the policy of "ecological civilization" was incorporated into the five-year national development plan for the first time [97]. China has already completed the 2020 goal of addressing climate change and establishing nature reserves ahead of schedule [98]. At present, biodiversity loss and ecosystem degradation pose major risks to the sustainable development of many countries, including China. According to the national development plan, it is of great significance to protect biodiversity and promote sustainable development.

Urban ecosystems are threatened by the process of increasing the density of buildings. More than 30% of occurring disease events were due to land-use change, agricultural expansion, and urbanization. Biodiversity loss, climate change, and the emergence of pandemics are the result of global environmental changes caused by unsustainable consumption. During these COVID-19 pandemic times, we need more scientific data to make the right policy choices and prevent future epidemics. It also benefits human health, biodiversity conservation, the economy, and sustainable development [99].

In many of the mega cities of the world, air quality deterioration is a serious environmental problem. For example, there is a report that 21,000 premature deaths are due to air pollution each year in Canada [100]. Urban parks with trees can play a significant role in reducing air pollution [101]. For example, turban trees are reported to remove 651,000 tons of pollutants in the air per year in the United States [102]. It was economically viable to improve air quality by means of urban forest management in Santiago, Chile [103]. Therefore, urban designers should consider the role of urban forests in improving the urban environment with their ecosystem services, especially air quality regulation in such populated cities as Beijing.

**5. Conclusions**

We found that air quality regulation ES and fresh water provision ES were considered the most important services for Beijing residents in terms of their choices of urban forest management strategies. In addition, Beijing residents are willing to pay 0.59% to 1.84% of their average monthly household income for urban forests expansion in order to improve

air quality annually. Citizens living in apartment and with high income are more willing to pay for various ESs from urban forests. Specifically, apartment owners are willing to pay more for urban forest ESs than those who do not own an apartment. Residents were more willing to pay for urban forest ESs as their income increases. We conclude that Beijing citizens are willing to pay more municipality tax in order to support urban forestry for air quality improvement. Based on these findings, we suggest that urban environmental policy makers should pay more attention to the ESs of forests (especially regulation function improving air quality) when designing and managing urban forests. In this way, the general public should be invited to participate in the governance of urban land use and ecosystem management so that they can cooperate with the government and other stakeholders in the environmental management. This study provides some experience for understanding resident's preference of urban forest ecosystem services and provides some basis and solution for urban forestry planning and management.

**Author Contributions:** H.Z.-Y. conceived, designed the study, conducted the survey, analyzed the data, interpreted the results, and wrote the first manuscript. Y.Y.-C. supervised the overall research, interpreted the results, and reviewed and edited the draft manuscript. All authors have read and agreed to the published version of the manuscript.

**Funding:** This research was carried out with the support of Seoul National University Carbon Sink Graduate Program provided by the Korea Forest Service & Korea Forestry Promotion Institute.

**Acknowledgments:** We are grateful to the carbon sink graduate program from the Korea Forest Service (KFS), Seoul National University Global Scholarship and BK21 PLUS (Brain Korea 21 Program for Leading Universities & Students) for granting the scholarship for the Ph.D study for the first author. We are also grateful to 30 experts for sharing their knowledge and opinions with questionnaires. We are also thankful for the efforts of our wonderful survey team receiving the available data. Lastly, our great gratitude is extended to all respondents of the survey for sharing their preferences and choices with us.

**Conflicts of Interest:** The authors declare no conflict of interest. The funders had no role in the design of the study; in the collection, analyses, or interpretation of data; in the writing of the manuscript, or in the decision to publish the results.

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
