# Peer review of "Beijing Resident’s Preferences of Ecosystem Services of Urban Forests"

_forests, doi:10.3390/f12010014_

Round 1

Reviewer 1 Report

Dear Authors, thank you very much for your very interesting read and very nice work. Nonetheless, it would be worthwhile to have some small revisions and a bit more work spent on working on the results and discussion section.

Methods:

Literature review: It is before the method section though you depend on the literature study for two purposes – identifying ES for your study and for the description on the situation on the state in Beijing. For that purpose, I´d suggest to reorganize your paper a bit to have a short description of the literature review for identifying the ES in the methods section. For this purpose, it would be good to add a few lines how you conduct your literature with a few lines e.g. with data banks such as Web of Science or other source such as lists of ES you used to identify the literature you considered for the paper and (pre-)selection of ES out of those huge catalogues.  Although it is quite common set you come up with, a short description how and based on which approach would be nice to have it better understandable and comprehendible.

A second point is the results and the discussion section. It can be done like this in combination though personally, I prefer to clearly separate between results presenting the findings and a discussion section where discussing them. I leave it to the authors or follow other reviewers if they comment on this as you have several approaches and separation into two chapters might be then difficult to read and I understand the intention of the authors to present it this chosen way. I´d leave it up to the authors and their response on it or other reviewers.

In this result and discussion section, I miss discussing the findings in a broader context and drawing some comparisons with other ES studies. Adding some reflections on it would be very interesting to discuss and compare your result in a broader context involving some literature – e.g. in relation of the high ranking of regulation functions and especially air quality. Is it something similar for other big cities and challenges that every city has to deal with? Though in principle your stand-alone results are very interesting, I strongly recommend adding some sections to put your work into the broader context and, for the conclusion sections, what can be lessons learned from your study in Beijing beyond your case and for other cities. In regards to the methods used, what would be the take-home message from your chosen approaches?

A minor point:

It might be nice to include a simple map or illustration showing the greenspaces and urban forests to get an overall idea of green space and urban forest distribution in Beijing for those not being so familiar with the city.

Reviewer 2 Report

The paper covers a very relevant topic in urban forest planning. The research approach is sound and  the paper is very well structured. The review of literature could perhaps be expanded in regard to  similar studies and their results. In the results and conclusions I would express the willingness to pay as a percentage of average income rather than in monetary terms, not comparable between countries.
